# A new dimension for magnetosensitive e-skins: active matrix integrated micro-origami sensor arrays

Christian Becker[1,2,3,5], Bin Bao [2,5✉], Dmitriy D. Karnaushenko[1,2,3], Vineeth Kumar Bandari[1,3], Boris Rivkin [2], Zhe Li[1,3], Maryam Faghih[2], Daniil Karnaushenko[1,2,3✉] & Oliver G. Schmidt [1,2,3,4✉]

Magnetic sensors are widely used in our daily life for assessing the position and orientation of objects. Recently, the magnetic sensing modality has been introduced to electronic skins (e-skins), enabling remote perception of moving objects. However, the integration density of magnetic sensors is limited and the vector properties of the magnetic field cannot be fully explored since the sensors can only perceive field components in one or two dimensions. Here, we report an approach to fabricate high-density integrated active matrix magnetic sensor with three-dimensional (3D) magnetic vector field sensing capability. The 3D magnetic sensor is composed of an array of self-assembled micro-origami cubic architectures with biased anisotropic magnetoresistance (AMR) sensors manufactured in a wafer-scale process. Integrating the 3D magnetic sensors into an e-skin with embedded magnetic hairs enables real-time multidirectional tactile perception. We demonstrate a versatile approach for the fabrication of active matrix integrated 3D sensor arrays using micro-origami and pave the way for new electronic devices relying on the autonomous rearrangement of functional elements in space.

[1] Center for Materials, Architectures and Integration of Nanomembranes (MAIN), Chemnitz University of Technology, 09126 Chemnitz, Germany. [2] Institute for Integrative Nanosciences, Leibniz IFW Dresden, 01069 Dresden, Germany. [3] Material Systems for Nanoelectronics, Chemnitz University of Technology, 09107 Chemnitz, Germany. [4] Nanophysics, Faculty of Physics, TU Dresden, 01062 Dresden, Germany. [5]These authors contributed equally: Christian Becker, Bin Bao. ✉email: b.bao@ifw-dresden.de; daniil.karnaushenko@main.tu-chemnitz.de; oliver.schmidt@main.tu-chemnitz.de

ntegrating high-density sensors into active matrix circuits is a vital research field that will enable future artificial e-skins with various receptors needed in emerging robotics, prosthetics and health monitoring devices[1,2]. These sensors can perceive temperature[3], humidity[4], illumination[5] and tactile stimuli such as pressure[6], strain[7] and shear forces[8]. Recently, various thin-film magnetic sensors such as planar Hall sensors[9], giant magnetoresistance (GMR)[10] and AMR sensors[11,12] have been integrated into e-skins to retool them with a magnetic sense for remote perception of magnetic objects and their motions. By incorporating magnetic nanocomposite cilia into elastic artificial skins, external mechanical stimuli can be transduced with high sensitivity through the detection of the change of magnetic stray fields[13]. However, state-of-the-art magnetic e-skins can only sense in one or two dimensions at small spatial resolutions, which constrains their application in the detection of the 3D magnetic vector field. To fully explore complex 3D magnetic vector fields and track magnetic objects in 3D space, three-axis magnetic sensors with high pixel density are desired. Conventional 3D Hall sensors can be fabricated with high pixel density on silicon chips, but their sensitivity is limited compared to magnetoresistance sensors[14,15]. As it is challenging to integrate highly sensitive 3D magnetoresistance sensors into large-area, high-density active matrices by mainstream planar microfabrication technologies, new integration strategies that are complementary to conventional microfabrication are required.

Recently, origami-inspired self-assembly routes have emerged as a paradigm for building 3D architectures with various functionalities[16–19], creating microelectronic devices such as sensors[20,21], actuators[22] and robotic systems[23]. The micro-origami approaches can preserve the advantages of photolithography in planar material deployment while rearranging the parallely obtained functional elements in 3D space with high precision[24]. Therefore, these approaches have unprecedented potential in miniaturization and integration of 3D sensor devices with high pixel density. Single devices based on different micro-origami processes have been explored by a few groups. For example, single cell gripper arrays and self-folded 2D material membrane microstructures have been demonstrated[25–27]. Actuators with embedded electronics have been fabricated by combining self-folding with 3D printing[28]. Although progress has been made, the integration level of devices is not sufficient. Recently, highly integrated 3D sensors with pixel arrays such as photodetector-based electronic eye cameras[29] and piezoresistive structure-based multimodal microelectromechanical sensors[30] have been demonstrated based on passive matrix pixels. Different from a passive architecture, active matrix technologies are used for driving sensor arrays to achieve high-density integration by dramatically decreasing the cable connections and signal crosstalk[31,32]. However, it is challenging to integrate active matrix driving circuitry with a self-assembled micro-origami 3D sensor array on one chip as the individual fabrication steps have to be compatible with each other.

In this article, we report an approach to fabricate active matrix integrated micro-origami magnetic sensor (IMOS) devices that can be used for spatiotemporal decoding of 3D magnetic fields. A multilayer polymeric platform is designed to realize self-folded micro-origami cubic architectures. Amorphous indium-gallium-zinc oxide thin-film transistors (a-IGZO TFTs) are integrated as the active matrix backplane and AMR sensors based on a NiFe alloy, namely permalloy (Py), are embedded into the self-folding polymer stack as sensing elements. The integrated electronics are fully compatible with the self-folding micro-origami process. The IMOS device is capable of sensing the magnetic field with high spatial resolution due to the high-density integrated sensor pixels (pitch 1.1 mm × 1.1 mm) with three orthogonally oriented subpixels. Static magnetic vector field mapping and real-time tracking of moving magnetic objects are measured and compared with simulation results. The IMOS device is integrated into an e-skin with embedded magnetic hairs for real-time tactile perception (Fig. 1a). A subtle touch of the artificial hair translates into the motion of a magnetic hair bulb loosely fixed within the elastomeric layer, which modulates the magnetic vector field in the IMOS device plane, thus transducing the direction of the touch. The allocation of 3D reshaped magneto-sensory pixels within the IMOS array allows to detect the distributed magnetic stray fields of these loosely fixed magnets as well as allowing a convenient information readout approach. Artificial magnetic hair integrated e-skins will find potential applications in robotics and soft robotics for direction-resolved touch sensation, pressure sensing, air and fluid flow detections[33–35]. The IMOS devices are not only limited to magnetic perception but serve as a platform that can be extended for instance to photonic and ultrasound detection and emission[36–38]. Combining active matrix circuits with self-assembled micro-origami devices would be a general strategy for the fabrication and high-density integration of 3D sensors which will find numerous applications in e-skins, soft robotics and related fields.

## Results

**Self-foldable polymeric platform with integrated electronics.** The design of a self-foldable polymeric platform is essential for the fabrication of the IMOS devices[39–41]. The IMOS devices are realized by autonomous folding of photolithographically defined planar microstructures into cubic architectures in this way rearranging functional elements in 3D space. The self-foldable polymer stack interconnects adjacent sides of the "cube" in order to couple the folding in such a way that orthogonally aligned principal planes are obtained (Fig. 1b). An a-IGZO active matrix is on-chip integrated for pixel-selective power supply and signal readout, allowing a high-density integration of sensors. The a-IGZO backplane is passivated by a polyimide (PI) layer. The self-foldable polymeric platform with the embedded sensors is constructed on top of the passivation layer (Fig. 1c). AMR sensors are employed to demonstrate a specific application of the IMOS concept in magnetic sensation. The embedded sensors and a-IGZO backplane are interconnected through via holes. The key polymer stack for the autonomous folding includes the sacrificial layer (SL), swelling hydrogel (HG), supporting PI and stiffening SU-8 layers (Fig. 1c). The SL is first etched to partially release the subsequent layers, then self-folding is initiated by swelling of the HG hinges which causes tensile stress at the interface to the supporting PI, providing sufficient actuation force to complete the autonomous folding process. Thanks to the coupled sides, accurate and identical orthogonally folded cubes can be achieved without any tedious adjustment of the induced interfacial stress. The PI supporting layer is partially anchored to the passivation layer and acts as a substrate to carry the sensors and interconnects, meanwhile providing the actuation force in the specific hinge areas. SU-8 photoresist (~10 μm) is patterned to stiffen the PI except the hinge areas to guarantee local bending and keep the cube sides planar. The SU-8 layer also functions as a passivation for the magnetic sensors to isolate them from the solvents used during the self-folding process.

The self-folding process and the folding quality of the IMOS device are studied as shown in Supplementary Fig. 1. Supplementary Fig. 1a demonstrates the self-folding process of a typical pixel in the device. To benchmark the folding quality, the orientation angles between each pair of planes, namely $\theta_1$ (XY and YZ), $\theta_2$ (YZ and XZ) and $\theta_3$ (XY and XZ) are measured (Supplementary Fig. 1b, c). The spatial and statistical

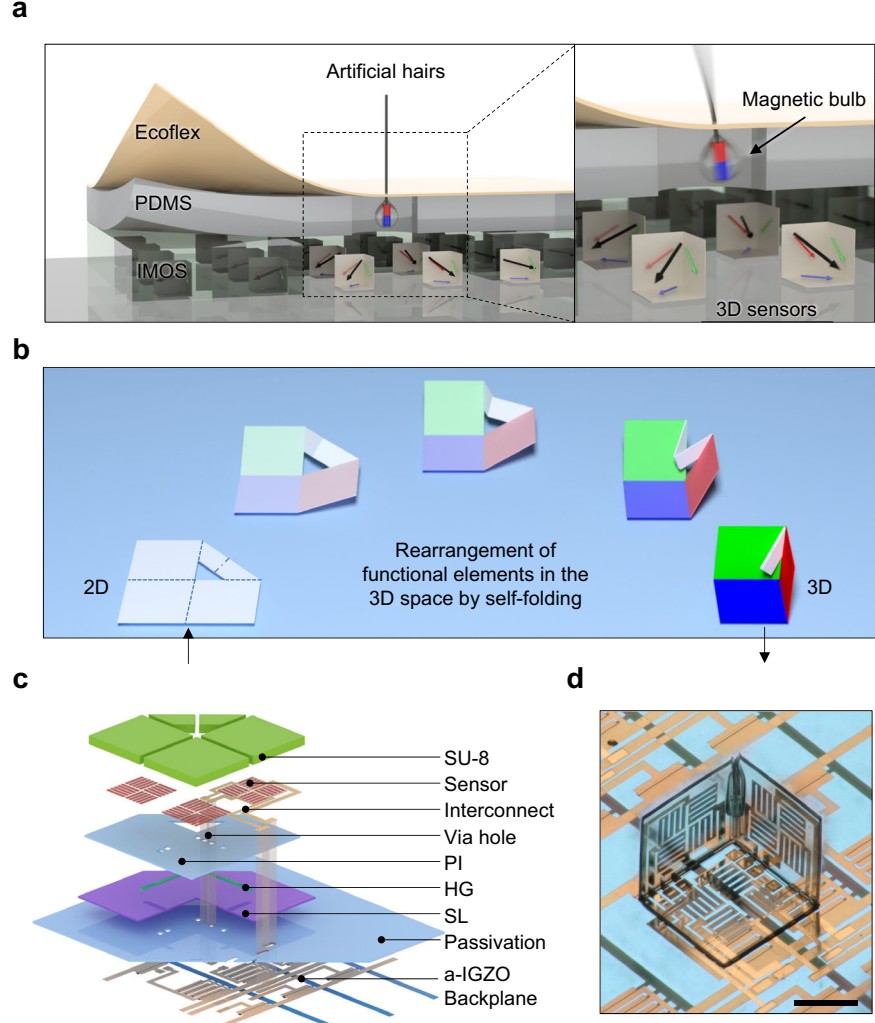

**Fig. 1 Concept and structure of the IMOS device. a** Vision of the IMOS devices for e-skin application. Conceptual images illustrating the application of the IMOS device in the hair-inspired sensation system for mechanoreception. The stray field strength and direction of the hair-attached small magnets would be changed due to bending of the hair, which is detectable by the IMOS device. **b** Deterministic self-folding process. Illustration of the self-folding micro-origami process. Orthogonal folding is obtained due to the rational design of the layout and the hinges. **c** Layout design of the self-foldable platform. Exploded schematic of the layout for the self-foldable polymeric platform with embedded electronics and integrated active matrix driving circuitry. The embedded electronics and the a-IGZO active matrix backplane are interconnected through via holes. **d** One pixel in the 3D sensor. Image of a single pixel which is composed of a self-folded polymer cube containing three sensors located on the orthogonal planes. The side length of the cube is 400 μm. Scale bar, 200 μm.

distributions of the folding angles for a typical sensor device with 8 × 8 pixels are shown in Supplementary Fig. 1d–f. The average values for $\theta_1$, $\theta_2$ and $\theta_3$ are $(90.4 \pm 0.6)°$, $(87.1 \pm 1.7)°$ and $(93.3 \pm 1.1)°$, respectively. Although small deviations from the ideal 90° angle can be observed for $\theta_2$ and $\theta_3$, the overall distributions are very narrow indicating a good reproducibility for the self-folding process. The small deviations are possibly from the obstruction of the hinges, which can be further improved by decreasing the hinge size.

In preceding design studies folded cubic microstructures with side lengths from 800 μm down to 200 μm have been already successfully prepared. Therefore, further miniaturization of the IMOS devices is possible in regard to the polymeric platform and will be finally limited by the electronics that needs to be integrated on chip. The presented self-folding concept has enabled the realization of the 3D architecture through the 2D design before folding, while the involved interfacial stress plays a minor role in determining the final microstructures. Besides, the planes on the

cube sides can provide proper locations for deploying the sensing elements when considering their lateral dimensions.

**3D active matrix integrated magnetic sensor**. The fabrication of the IMOS devices consists of three major steps: (1) preparation of the a-IGZO active matrix backplane, (2) fabrication of the foldable polymer stack with embedded sensors, (3) and autonomous folding to rearrange the sensors (Supplementary Fig. 2a). The first two steps are fully compatible with conventional photolithographic processes, where established microfabrication tools such as photopatterning, physical and chemical vapor depositions, wet and dry etching can be employed[42,43]. Due to the parallel nature of the microfabrication techniques and the self-folding process, large-area arrays with many identical pixels can be obtained in the same fabrication batch. For demonstration purposes, we fabricate IMOS devices with an 8 × 8 pixel array and the pixel pitch is 1.1 mm × 1.1 mm (Fig. 2a, b, Supplementary Fig. 2b–f).

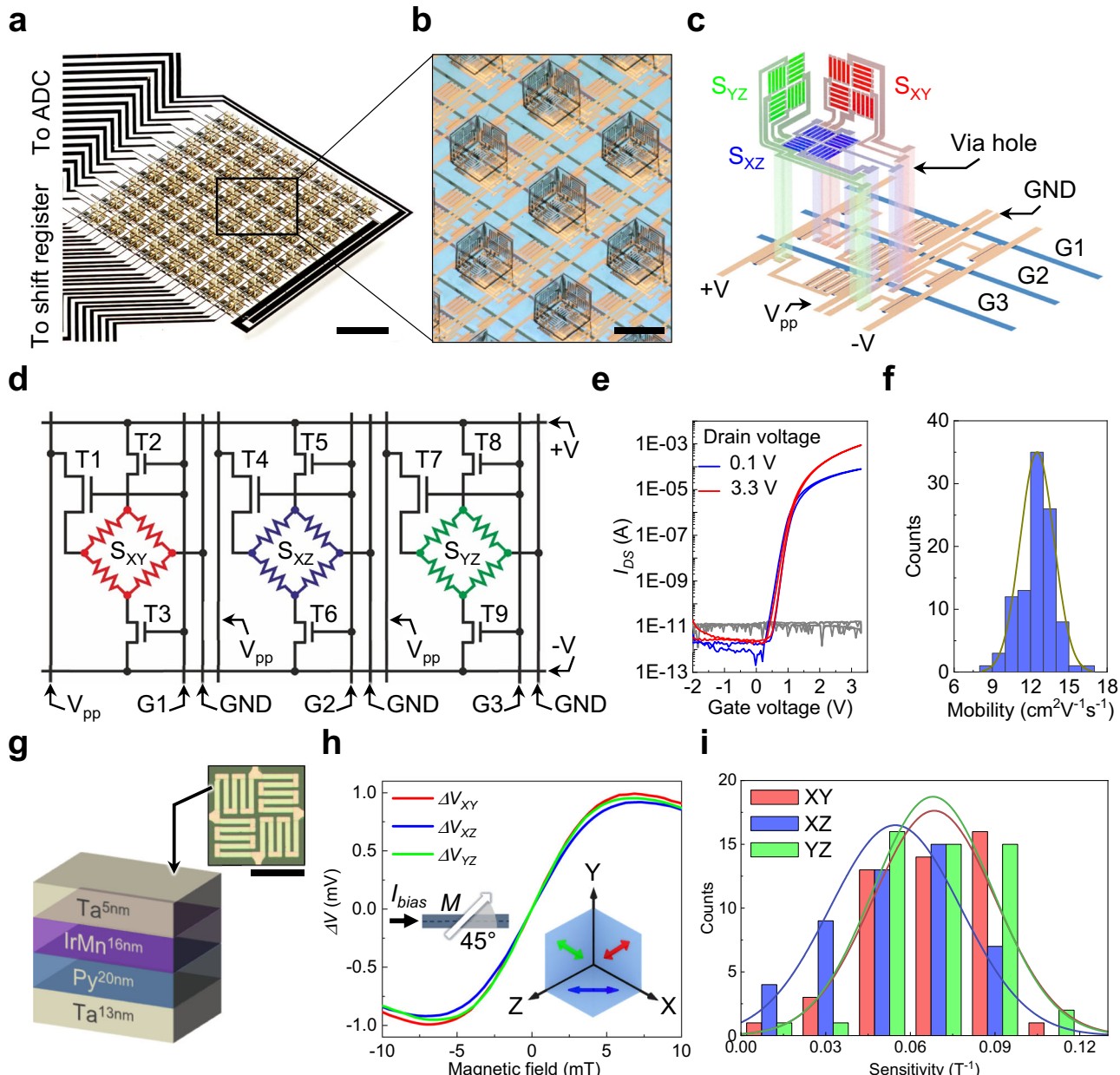

**Fig. 2 Structure and magnetoelectrical performance of the IMOS device. a** Photo of an IMOS device with 8 × 8 pixel matrix. Scale bar, 2 mm. **b** Image of the pixel array in the device. The pixel pitch is 1.1 mm × 1.1 mm. Scale bar, 500 μm. **c**, **d** Exploded schematic and circuit diagram of a single pixel which contains three subpixels. The Wheatstone bridge type AMR sensors on the three orthogonal planes are marked as $S_{XY}$, $S_{XZ}$ and $S_{YZ}$ respectively. **e** Typical transfer characteristics with $V_{GS}$ from -2 V to 3.3 V for the a-IGZO TFTs. The leakage currents are plotted in gray lines. **f** Statistics of the mobility $\mu$. Histogram of $\mu$ for 100 a-IGZO TFTs. The average $\mu$ is (12.51 ± 1.32) cm$^2$ V$^{-1}$ s$^{-1}$. **g** Schematic illustration of the layer stack of the AMR sensors. Inset is a micrograph of the AMR sensor. Scale bar indicates 200 μm. **h** Typical responses of the $S_{XY}$, $S_{XZ}$ and $S_{YZ}$ sensors. The left inset shows the 45° angle between the bias current and the magnetization direction. The right inset shows the orientations of the sensitivity directions of $S_{XY}$, $S_{XZ}$, and $S_{YZ}$. For all the tests $V_{pp} = V_G = 3.3$ V. **i** Histograms of the sensitivities for the working sensors on the XY, XZ, YZ planes in a typical 8 × 8 sensor array. The sensitivities are normalized to $V_{pp}$ of 3.3 V. Source data are provided as a Source Data file.

One pixel contains three magnetic bridges (subpixels) located on the orthogonal planes of the self-folded cube (Figs. 1d and 2c). Each subpixel has one Wheatstone bridge type AMR sensor, one driving TFT (dTFT) and two readout TFTs (rTFTs), which we define as the 3T1S design (Fig. 2d). The AMR sensors on the orthogonal planes are marked as $S_{XY}$, $S_{XZ}$ and $S_{YZ}$ (Fig. 2c). T1, T4 and T7 are the dTFTs, and T2, T3, T5, T6, T8 and T9 are the rTFTs (Fig. 2d). The TFTs in one subpixel share the same gate line thus can be activated synchronously. For instance, when T1, T2, and T3 are switched on through gate line 1 (G1), bias current

flows through the $V_{PP}$ line to the channel of T1 and then to $S_{XY}$ and the ground line (GND). Differential voltage is picked up through T2, T3 and the +V, −V lines leading to the external measurement electronics. Due to this design, the subpixels on the 24 gate lines can be scanned serially and the output voltage can be parallelly read out by the 8-channel analog to digital converter (ADC), which guarantees a fast response of the 8 × 8 matrix. The frame rate of our 8 × 8 sensor array can reach up to 24 Hz thus realizing real-time dynamic tracking of magnetic objects (Will be discussed later).

**Electrical and magnetoelectrical performance**. In our device a-IGZO TFTs are used to construct the active matrix backplane due to their high mobility, large-area uniformity and low processing temperature[44–46]. Only one geometry of a-IGZO TFTs ($L = 3\,\mu m$, $W = 200\,\mu m$) is used to simplify the design of the backplane. The rTFTs (T2 and T3) are composed of single a-IGZO TFTs and the dTFTs (T1) are constructed by four a-IGZO TFTs in parallel (Supplementary Fig. 3a, b). The large size of the dTFTs can provide a considerable bias current (~ 3 mA at $V_{GS} = V_{DS} = 3.3$ V, Supplementary Fig. 3g), thus maintaining high sensitivity of the AMR sensors. Meanwhile, a high-k dielectric layer stack of $(HfO_2^{2.5\,nm}/Al_2O_3^{2.3\,nm})_3/HfO_2^{2.5\,nm}$ with an areal dielectric constant of $4.9 \times 10^{-7}$ F/cm$^2$ is used to decrease the operation voltage down to 3.3 V[46]. Typical transfer and output characteristics show that our n-type a-IGZO TFTs work in the enhancement mode with an on/off ratio up to $10^8$ (Fig. 2e and Supplementary Fig. 3c–h). The electrical performance of the a-IGZO TFTs combined with the special 3T1S subpixel design can greatly decrease the signal crosstalk. The narrow distribution of the mobilities ($\mu$) for 100 TFTs documents the uniformity of the fabrication (Fig. 2f). A good stability of the a-IGZO TFTs during the self-folding process is essential for their integration into the 3D sensor arrays, hence the statistics of the TFT performance before and after the self-folding is studied (Supplementary Fig. 4). Before the self-folding, the on/off ratio is $(1.08 \pm 0.18) \times 10^8$, the threshold voltage ($V_{th}$) is $(0.95 \pm 0.13)$ V, the subthreshold swing ($SS$) is $(154 \pm 17)$ mV/dec, and the $\mu$ is $(12.51 \pm 1.32)$ cm$^2$ V$^{-1}$ s$^{-1}$. After the folding process the on/off ratio is $(1.09 \pm 0.42) \times 10^8$, the $V_{th}$ is $(1.02 \pm 0.10)$ V, the $SS$ is $(157 \pm 15)$ mV/dec, and the $\mu$ is $(12.16 \pm 1.19)$ cm$^2$ V$^{-1}$ s$^{-1}$. Therefore, there is no significant change of the electrical performance for the a-IGZO TFTs, indicating that the a-IGZO active matrix is compatible with the self-folding process.

In our proof-of-concept demonstration, AMR sensor elements are deployed to construct 3D magnetic sensor arrays because of their high sensitivity and linear response in the vicinity of zero field[47–49]. The layer stack sequence of the AMR sensor is Ta$^{13\,nm}$/Py$^{20\,nm}$/IrMn$^{16\,nm}$/Ta$^{5\,nm}$ (Fig. 2g). A Ta seed layer is used to induce the desired crystal orientation in the antiferromagnetic (AFM) IrMn layer, and the top Ta serves as capping layer to prevent metal oxidation. The AFM layer is utilized to define the zero-field magnetization direction. Py is used for sensing external magnetic fields based on anisotropic magnetoresistance which is defined by $R = R_\parallel - R_{max} \sin^2\theta_{jM}$, with $R_{max} = R_\parallel - R_\perp$ where $R_\parallel$ is the resistance when the magnetization is parallel to the bias current, $R_\perp$ is the resistance when the magnetization and the bias current are perpendicular, and $\theta_{jM}$ is the angle between bias current and magnetization directions[11]. A Wheatstone bridge configuration is adopted to operate the sensor in a constant voltage mode and compensate for thermal drifts in the sensing elements (Inset of Fig. 2g and Supplementary Fig. 5a)[50,51]. The AMR stripe elements are interconnected by Au pads and rotated by 45° to the bias field during deposition, adjusting the sensor bridge to its maximum sensitivity (Left inset in Fig. 2h and Supplementary Fig. 5a). The typical response of the AMR sensor element on the Si substrate shows that the output voltage can reach + / − 6 mV with a constant bias of 1 V (Supplementary Fig. 5b). In the IMOS devices the electrical connection of the Wheatstone bridges leads to an inverted signal for $S_{XZ}$ measured along the sensitivity direction in the planar state (Supplementary Fig. 6). Therefore, in the acquisition software the $S_{XZ}$ response is inverted back to record the real polarity of the magnetic field. Typical responses of the $S_{XY}$, $S_{XZ}$ and $S_{YZ}$ in the folded 3D magnetic sensor pixels are shown in Fig. 2h. Although the output

signal drops to around 2 mV when the sensor elements are integrated into the IMOS devices due to the serial connection of a dTFT with the sensing bridge, the field strength down to 20.3 μT can still be unambiguously detected (Supplementary Fig. 8)[20]. The right inset in Fig. 2h displays the orientations of the sensitivity directions for $S_{XY}$, $S_{XZ}$ and $S_{YZ}$ in the Cartesian coordinate system by considering the initial magnetization direction and the folding process. The response of a single pixel to the field swept along six typical directions are recorded in Supplementary Fig. 7. The sweeping directions are along the sensitivity directions (Supplementary Fig. 7a, c, and e) and perpendicular to the sensing planes of $S_{XY}$, $S_{XZ}$ and $S_{YZ}$ (Supplementary Fig. 7b, d, and f), respectively. In one measurement, $S_{XY}$, $S_{XZ}$ and $S_{YZ}$ show different shapes of the response curves depending on their spatial orientations in the field.

The stability of the integrated sensors during the micro-origami process is a prerequisite for constructing IMOS devices by this approach. Thus, two key figure-of-merit parameters, yield and sensitivity, are compared before and after the self-folding process. The yields for $S_{XY}$, $S_{XZ}$ and $S_{YZ}$ sensor arrays before self-folding are 75.0%, 76.6% and 81.3% respectively, which slightly decrease to 75.0%, 75.0% and 78.3% after the folding (Supplementary Fig. 9a–c, g–i). The overall yield for all the sensor pixels drops from 77.6% to 76.1% due to the folding process. Therefore, the self-folding process has only little influence on the yield of the sensors. The yield is mainly limited by the multilayer photo-lithographic processes which can be further improved by optimized pixel circuit design and advanced clean-room facilities. Figure 2i summarizes the statistical distributions of the sensitivities for the working sensors on the XY, XZ and YZ planes in the folded state. The average sensitivities are $(0.068 \pm 0.022)$ T$^{-1}$, $(0.055 \pm 0.023)$ T$^{-1}$ and $(0.068 \pm 0.021)$ T$^{-1}$, respectively. Before the folding process these values are $(0.064 \pm 0.018)$ T$^{-1}$, $(0.054 \pm 0.020)$ T$^{-1}$ and $(0.059 \pm 0.017)$ T$^{-1}$, respectively (Supplementary Fig. 9d–f, j–l). The sensitivity distribution is relatively large compared to the narrow distribution of the TFT mobilities (Fig. 2f), which is probably due to the electrical resistance variations of the wires and vias in the circuits, especially the vias connecting different conducting layers. The sensitivity distribution can be further improved by via filling with additional depositions of metal layers. From the change of the sensitivities before and after folding one can see that the self-folding process has also negligible effect on the sensitivity of the 3D magnetic sensors. Therefore, the micro-origami self-folding approach can safely serve as a general procedure for the 3D sensor fabrication thanks to its minor influence on the performance of the integrated electronics[52].

Besides the stability of the integrated electronics during the micro-origami process, the operational stabilities of the a-IGZO TFTs and the IMOS sensors are also studied as shown in Supplementary 10. First, constant-voltage-bias-stress test is performed to investigate the stability of the a-IGZO TFTs (Supplementary Fig. 10a, b). During the stress over a duration of 3600 s, the drain current drops within one order of magnitude and the leakage current dose not increase (Supplementary Fig. 10a). From the transfer curves before and after the stress test we can see that the a-IGZO TFT is still functioning well after the stress process, revealing a good stability of the TFTs under operation (Supplementary Fig. 10b). The dynamic stability of the IMOS device has been tested by continuously recording the response of a single pixel over 3 min while approaching a small magnet from time to time. Clear response spikes can be seen while the baselines for all the three sensors in the pixel only show a small drift, indicating the high operational stability of the IMOS device (Supplementary Fig. 10c).

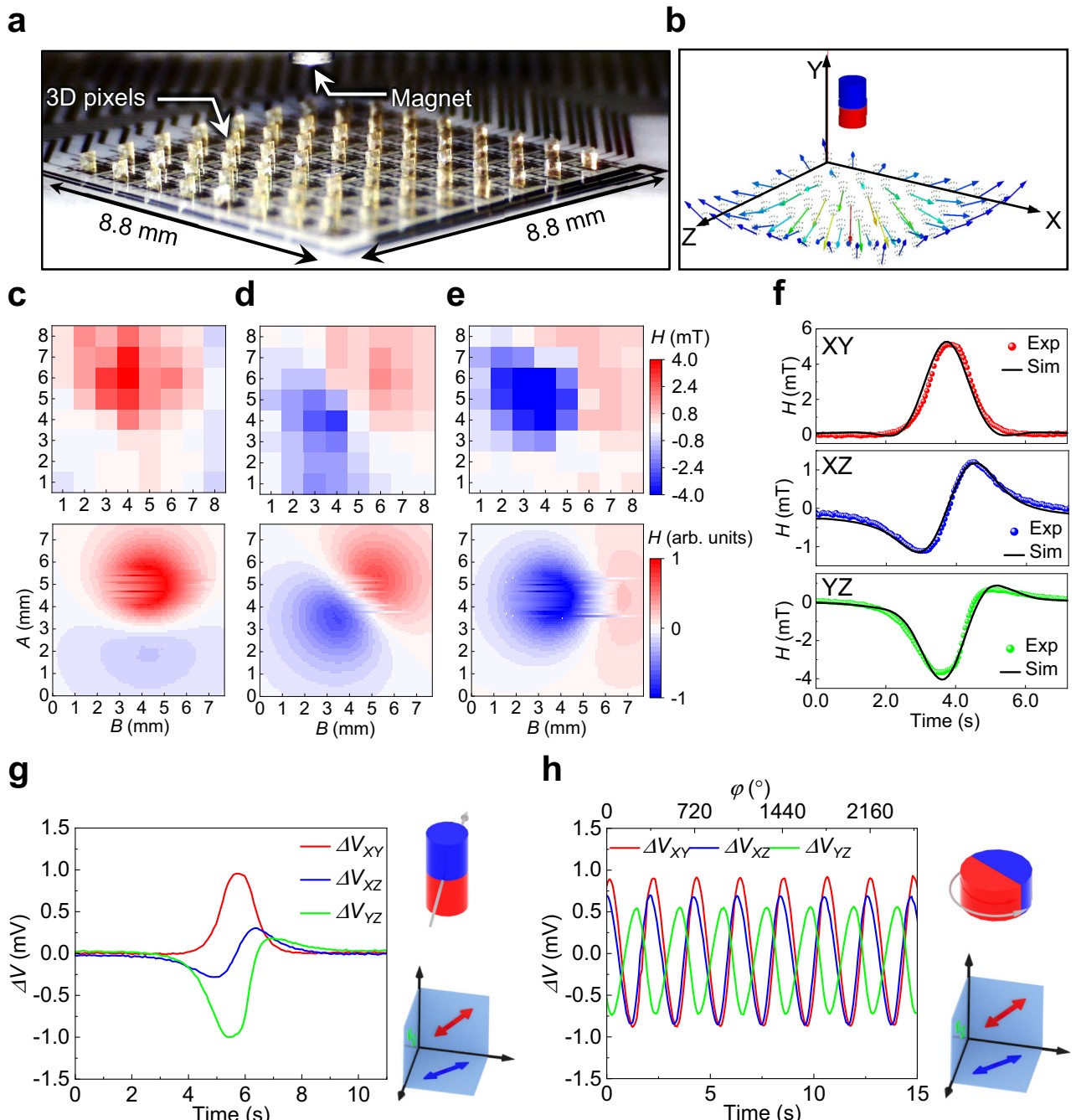

**Fig. 3 Static mapping and dynamic tracking of magnetic objects. a** Image of an IMOS device together with a small NdFeB permanent magnet. **b** Simulation result of the magnetic vectors at the locations of the 8 × 8 sensor pixels. **c–e** Static mapping of the strengths of the magnetic stray field by $S_{XY}$, $S_{XZ}$ and $S_{YZ}$ sensor arrays on the XY, XZ and YZ planes, respectively. The top panel is from the experimental results and the bottom panel is from simulation. The magnet and its relative position to the sensor array is shown in **a**. **f** Simulation result of the dynamic responses of $S_{XY}$, $S_{XZ}$ and $S_{YZ}$ in a single pixel to the linear motion of a magnet. The dotted lines are measured results and the continuous lines are from simulation. **g** Dynamic voltage response to a linear motion of the NdFeB permanent magnet monitored by a single pixel. **h** Dynamic tracking of the rotation of a NdFeB permanent magnet by a single pixel. The output voltages for $S_{XY}$, $S_{XZ}$ and $S_{YZ}$ change periodically with constant magnitudes emphasizing the stability of the sensors. The 90° phase shift between the signals from $S_{XY}$ and $S_{YZ}$ indicates the orthogonal folding of these two sensing planes. The relative positions of the magnets to the sensor pixels as well as the magnetization directions of the magnets are schematically illustrated on the right of the response curves. Source data are provided as a Source Data file.

**Static mapping and dynamic tracking of magnetic objects.** Static mapping and dynamic tracking of magnetic objects in 3D space is of significance for various applications such as the localization and navigation of microrobots[53] as well as the feedback control for soft robotics[54]. The static mapping of the stray field of a small NdFeB permanent magnet is performed by placing the magnet over the IMOS device and recording the output voltages of the 3D sensor arrays (Fig. 3a). Mapping of the magnetic stray field strengths by the $S_{XY}$, $S_{XZ}$ and $S_{YZ}$ sensor arrays can be done after calibration of the IMOS device (Top panel in

Fig. 3c–e). Note that the field strengths for the dysfunctional pixels are interpolated from adjacent pixels. Each sensor plane measures the projection of the local magnetic stray field vector. For instance, in Fig. 3d the distribution of the magnetic stray field vector component parallel to the sample plane is imaged by the $S_{XZ}$ sensor arrays, revealing the dipole nature of the axially magnetized magnet. Figure 3c, e show the out-of-plane components imaged by the $S_{XY}$ and $S_{YZ}$ sensor arrays respectively, which in this case show similar response due to the symmetry of the magnetic stray field. Note that the responses for $S_{XY}$ and $S_{YZ}$ sensors are inverse due to the orientation of the sensitivity directions. The simulation result of the magnetic vector field orientation at the $8 \times 8$ pixels is shown in Fig. 3b. Based on these vector orientations the maps of the magnetic stray field components measured by the $S_{XY}$, $S_{XZ}$ and $S_{YZ}$ sensor arrays are calculated (Bottom panel in Fig. 3c–e). Our experimental maps match well with the simulation results, demonstrating the validity of the IMOS device in the static mapping of magnetic stray fields. The simulation model is discussed in the Methods and Supporting Information.

The IMOS device can also be used for dynamic tracking of the motion of magnetic objects in real time. The device is placed underneath a linear translational stage to which a small permanent magnet is attached and moved across the matrix. A typical dynamic response of a single pixel to the linear motion of the magnet is shown in Fig. 3g. A small delay of the $S_{XY}$ response with respect to the responses of $S_{XZ}$ and $S_{YZ}$ is noticeable. This reflects the spatial distribution of the individual subpixel sensors (Illustration in Fig. 3g). The magnet reaches the $S_{XZ}$ and $S_{YZ}$ panels first and after about 0.3 s later the $S_{XY}$ panel also responds. The signals of two adjacent pixels to the linear motion in Supplementary Fig. 11a show that the two pixels have similar response but with a time delay of about 0.5 s. Considering the pixel pitch of 1.1 mm the actual magnetic object velocity of around 2.2 mm/s can be detected. The dynamic response to the linear motion is also simulated in Fig. 3f. Our experimental response curves match well with the simulated dynamic responses, validating the capability of the sensor to track magnetic objects. Another type of magnetic object motion is monitored by rotating a diametrally magnetized permanent magnet on top of the IMOS device (Fig. 3h). Sinusoidal responses with constant amplitudes are observed, emphasizing the stability of the voltage outputs without signal drift. The 90° phase shift between the signals from $S_{XY}$ and $S_{YZ}$ indicates the accurate orthogonal folding. While moving the rotating magnet away from the sensor the signal amplitudes decay with constant phase shifts (Supplementary Fig. 11b). The response of the $8 \times 8$ sensor array to a linear motion of the magnet is recorded in Supplementary Fig. 11c. The snapshots of the voltage response at different time for the $S_{XY}$, $S_{XZ}$ and $S_{YZ}$ sensor arrays are plotted respectively. The frame rate of the array in this case is around 20 Hz. The real-time tracking of the motion for a stripe magnet is recorded in Supporting Video 1. Supporting Video 2 shows the response of the IMOS device to the 3D motion of an axially magnetized small magnet. The static mapping and dynamic tracking results demonstrate that our IMOS device can be used as a magnetic camera for the spatiotemporal measurement of a magnetic field (Supplementary movie 1 and 2).

**Magnetic hair embedded e-skin**. A potential application of the IMOS device for a mechanoreceptive e-skin is demonstrated by integrating the device with a flexible composite skin layer and embedded magnetic hairs, making the system capable of detecting external mechanic stimuli and their directions (Fig. 4a). Like its natural counterpart, the artificial hairs provide the skin with feelers that can perceive environmental stimuli and enhance sensation beyond direct skin indentation. The

system is made of an encapsulated IMOS device stacked with an elastic polymeric bilayer into which artificial magnetic hairs are incorporated (Figs. 1a, 4b and Methods). The magnetic hair bulbs are hold in cavities of a poly(dimethylsiloxane) (PDMS) layer by an ultra-flexible Ecoflex film formed on top of the PDMS (Fig. 4b), thus it can tilt easily in response to an external force applied to the hairs. An external stimulus, for instance bending the hair in one direction leads to a curvilinear motion of the magnetic bulb in the opposite direction, which in turn is detected by the IMOS device (Illustrations in Fig. 4c). The recorded sensing data show distinguishable signal traces of each subpixel in response to mechanic stimulus of a single hair in different directions (Fig. 4c II–V). Note that after combining the IMOS device with the flexible skin a magnetic background is stored and subtracted, resulting in a flat standby baseline (Fig. 4c I). The hair resets quickly after release, recovering the initial position of the magnetic bulb in the cavity. This results in a good reproducibility and stability of the proposed sensation system. Different output voltage amplitudes correspond to different mechanic stimulus intensities. Transducing actual applied forces will require in depth modeling of the mechanical system and needs to take into account the IMOS properties in terms of magnetic field strength and angular resolution, which will be subject of future work. Finally, the concept is extended to the IMOS response mapping for a multi-hair skin with a $2 \times 2$ artificial hair array, enabling the allocation of stimuli throughout the skin (Fig. 4d). Here, the locations (hairs) 1–4 are successively stimulated and perceived as observed by the four sensor pixels exclusively responding to the individual hair in their proximity. The restoring force originating from the ultra-flexible Ecoflex layer is sufficient to reset the initial (standby) state but small enough to translate a gentle touch of the hair shaft into a perceptible magnetic bulb motion. Since the magnets are allowed to move with the skin layer with many possible directions, the 3D magnetic sensor array could enable the detection of the mechanical motions of the attached hairs in various directions which would be not possible to realize with a pair of planar AMR sensors. The artificial hairs and their magnetic bulbs are relatively large thus can be scaled down further. Large-area e-skins with denser multi-hairs and flexible IMOS devices will be compliant to curved surfaces and can be integrated for robotic and soft robotic application scenarios.

## Discussion

In our demonstration, exchange biased AMR sensors are used as the sensing modality to validate the concept of rearranging functional elements in 3D space by a micro-origami self-folding process. In our specific sensor magnetization case each single pixel has a virtual sensitivity plane which is similar to the 2D case, tilted to the substrate plane. By adjusting the initial magnetization directions during the fabrication, 3D magnetic sensors with various configurations for the sensitivity directions can be achieved after the self-folding process. A general description of the sensitivity direction rearrangement by the self-folding process is schematically illustrated in Supplementary Fig. 13 and discussed in Supplementary Note 1. In this work, we intend to emphasize on the single-step deposition and magnetization of the magnetic sensors. By combining 2D sensor layout designs with multiple deposition and magnetization steps, 3D magnetic sensors with orthogonal sensitivity directions can be achieved (Supplementary Fig. 13e), which cannot be realized for planar 2D cases even with multiple deposition and magnetization steps. Therefore, the micro-origami self-folding process would be of significance for the fabrication of 3D sensors by avoiding various technological and performance complications.

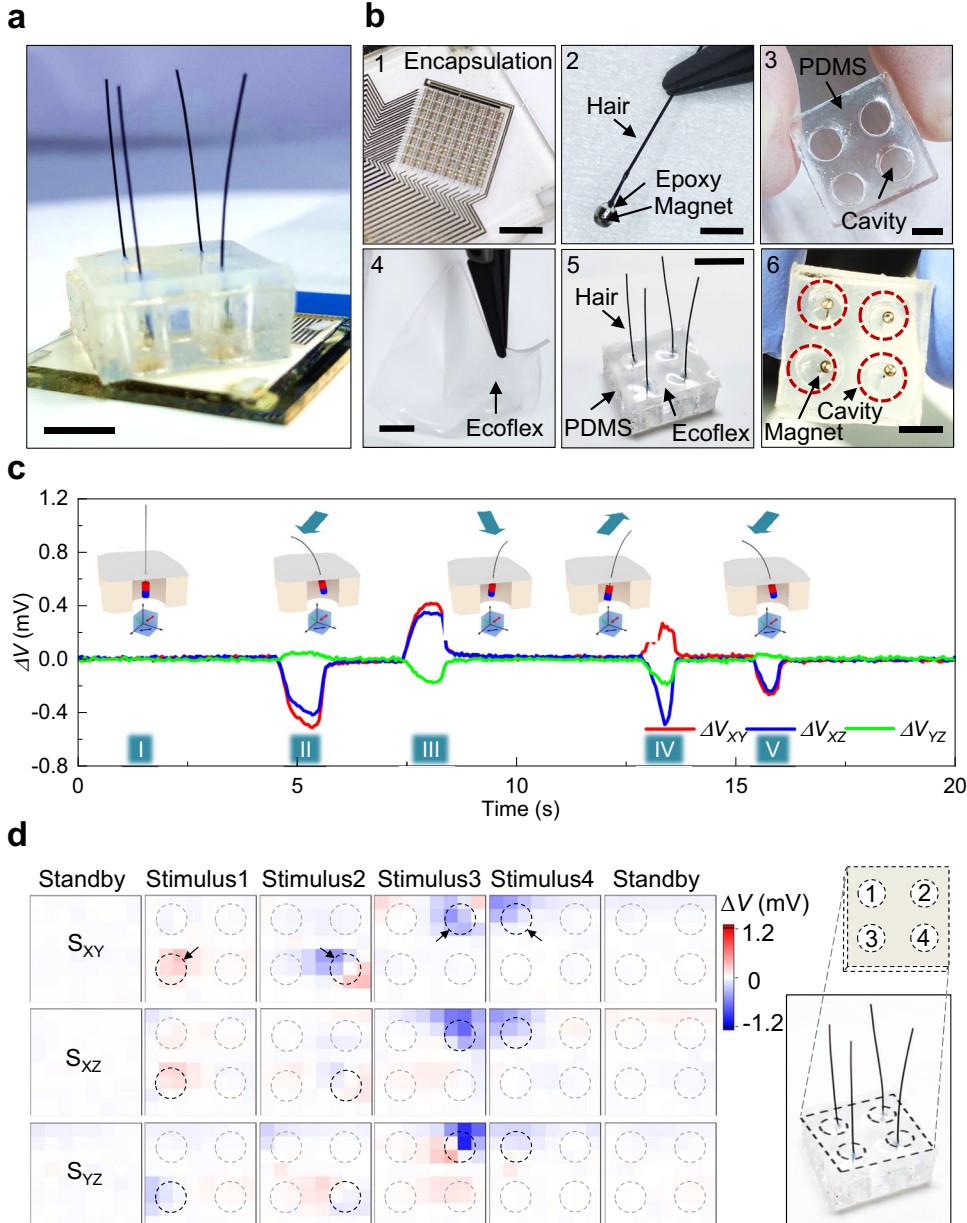

**Fig. 4 Magnetic hair embedded e-skin for mechanoreception. a** Photograph of the magnetic hair embedded e-skin system. It consists of an encapsulated IMOS array for signal reception and a magnetic hair embedded skin layer for the transducing of mechanic stimulus Scale bar, 5 mm. **b** Photographs of the integration process. 1, The IMOS device is encapsulated by epoxy. Scale bar, 4 mm. 2, The small magnet is adhered to a hair and embedded in epoxy. Scale bar, 2 mm. 3, A spacer PDMS film with cavities is fabricated. Scale bar, 5 mm. 4, Ultra-flexible Ecoflex film is casted. Scale bar, 2 mm. 5, Magnetic hair, PDMS spacer and Ecoflex skin are assembled. Scale bar, 15 mm. 6, Bottom view of the skin layer showing that the magnets are hold within the cavities of the PDMS spacer. Scale bar, 5 mm. **c** Single pixel response over time by bending the hair in different directions. The initial state (I) and the different bending directions (II–V) of the magnetic hair are illustrated schematically. **d** Response of the pixel array to the mechanical stimuli of multiple hairs. Source data are provided as a Source Data file.

In conclusion, we have demonstrated an approach to fabricate active matrix integrated micro-origami magnetic sensor device which can have 3D vector field sensing capability. The 3D magnetic sensor fabrication combines planar photolithography with self-folding micro-origami in a wafer-scale process, providing high-density integrated sensor pixels with three orthogonally oriented subpixel planes. The integrated IGZO active matrix backplane and the Py based AMR sensors are entirely compatible with the micro-origami approach. The 3D sensing features of the matrix is explored for the static mapping of magnetic stray fields and dynamic tracking of magnetic objects in real time. The 3D magnetic sensor is integrated into multi-hair embedded e-skin for real-time tactile perception with apparent use in soft robotics. By merging e-skin, micro-origami, active electronics and proper sensor technologies together, our approach frames a general strategy for the fabrication and high-density integration of 3D sensors for vector field detection far beyond the sensing of a magnetic field.

## Methods

**Fabrication of a-IGZO active matrix backplane.** Glasses with a dimension of 50 mm × 50 mm × 1 mm were used as substrates, which were thoroughly washed in a professional washer (Miele & Cie. KG, Gütersloh, Germany). Cr[5 nm]/Au[100 nm]/

Cr[20 nm] metal layers were subsequently deposited through e-beam evaporation (PLASSYS Bestek, Marolles-en-Hurepoix, France) and patterned by lift off as the gate layer. Dielectric layer of $(HfO_2^{2.5 nm}/Al_2O_3^{2.3 nm})_3/HfO_2^{2.5 nm}$ was deposited by plasma-enhanced atomic layer deposition (PEALD, FlexAL, Oxford Instruments plc, Abingdon, UK), and patterned by dry-etching with $CF_4$ and Ar gases (Plasma Lab 100, Oxford Instruments plc, Abingdon, UK). a-IGZO[15 nm] semiconductor layer was RF magnetron sputtered in an $Ar/O_2$ mixture atmosphere in a house-made sputtering machine (B2, IFW-Dresden, Dresden, Germany). The Ar flow rate was 30 sccm and $O_2$ flow rate was 0.5 sccm. The base pressure was $1 \times 10^{-7}$ mbar and the deposition pressure was $5 \times 10^{-3}$ mbar. The a-IGZO thin film was patterned by wet etching in a 4 wt% oxalic acid (Merck KGaA, Darmstadt, Germany) solution. Ti[70 nm]/Au[50 nm] was DC magnetron sputtered (DCA, IFW-Dresden, Dresden, Germany) and patterned by lift off as the source-drain layer. Finally, a 500-nm-thick photopatternable PI layer was spin-coated and lithographically patterned as the passivation layer, leaving the via holes and contact pads exposed to connect with the upper metal layers.

**Patterning of self-foldable polymer stack.** The origami-inspired autonomous folding approach was based on a fully photopatternable stimuli-responsive polymeric platform[20,23,55,56]. Before constructing the polymer stack on top of the PI-passivated a-IGZO active matrix backplane, the surface was treated with $O_2$ plasma (Pico, Diener electronic GmbH, Ebhausen, Germany) for 45 s to enhance the adhesion. Then, a lanthanum-acrylic acid-based organometallic photopatternable complex was spin-coated at 2000 revolutions per minute (*rpm*) and patterned on MA6 Mask Aligner (Karl Suss KG-Gmbh & Co, Munich-Garching, Germany) to form a 280-nm-thick SL layer. The photopatternable HG and PI were spin-coated and lithographically patterned using similar processes. The spin-coating rates were controlled to achieve a thickness of around 500 nm for both of the layers. The thicknesses of the polymer layers were measured on a profilometer (Bruker, Billerica, MA, USA) and the spin-coating rates were calibrated accordingly. In our case the spin-coating rate was 6000 rpm for HG, and 8000 rpm for PI, respectively.

**Deposition and patterning of magnetic sensor layers.** The AMR sensor elements were deposited and patterned directly on the surface of the polymers with the following layer sequence: Ta[13 nm]/Py[20 nm]/IrMn[16 nm]/Ta[5 nm]. The sensor stacks were prepared in Ar atmosphere by DC magnetron sputtering (B8, IFW-Dresden, Dresden, Germany) with a base pressure of $2.4 \times 10^{-6}$ mbar and a deposition pressure of $1.4 \times 10^{-3}$ mbar. The deposition power was kept as 100 W for Py, and 50 W for both Ta and IrMn. The thickness was adjusted by precisely controlling the time of the deposition sequence. Note that during deposition the sample was fixed on a magnetic holder with a bias field of around 25 mT and the AMR strips were aligned to be 45° to the bias field (Supplementary Fig. 5). The AMR strips were patterned by lift off. Then, a 70-nm-thick Au layer was DC sputtered (DCA, IFW-Dresden, Dresden, Germany) and patterned by lift off to connect the AMR strips, forming Wheatstone bridge type AMR sensor arrays. The Au layer also connected the sensor bridge arrays to the underneath a-IGZO active matrix backplane through the via holes. Finally, a 10-μm-thick SU-8 layer was patterned as a passivation layer for the sensors as well as a stiffening support to fix the sensing planes during the self-folding process. Note that the post-bake of the SU-8 layer was carried out at just 150 °C for 5 min to prevent demagnetization of the sensor elements.

**Device cable bonding.** The IMOS device was connected to a custom-made test printed circuit board (PCB) via a flexible cable (RS Components GmbH, Frankfurt, Germany) (Supplementary Fig. 12). The flexible cable was bonded to the contact pads of the sensor array through an anisotropic conductive film (ACF, 3 M 7371, St. Paul, MN, USA). The bonding was performed on a home-made ACF bonding device by applying a pressure of around 10 kg/cm$^2$ at 250 °C for 5 min.

**Micro-origami self-folding process.** The self-folding process began with the etching of the SL layer. The bonded device with planar sensor microstructures was immersed into a mixture solution of hypophosphoric acid and perchloric acid (SigmaAldrich Chemie GmbH, Taufkirchen, Germany) with pH around 2. After 20 min the SL layer was totally dissolved and the polymer stack on top of the SL layer were released from the substrates. Then the device was rinsed by deionized water and transferred to an alkaline solution of tetramethyl ammonium hydroxide (Micro Chem, MA, USA) with pH around 10. The HG was swollen and self-folding took place until the two sensing planes formed orthogonal folding (about 30 min). The device was taken out from the folding solution and dried on a hotplate at 60 °C for overnight. The IGZO TFT test arrays were also treated with the same processes to check their stability during the self-folding process.

**Characterization of the electrical performance of the a-IGZO TFTs.** The transfer and output characteristics of the a-IGZO TFT test structures were measured on an automatic probe station (Summit 12000, Cascade Microtech Inc., Beaverton, OR, USA) which was connected to a Precision Source/Measure Unit (B2902A, Agilent Technologies Inc., Santa Clara, CA, USA). The capacitance of the

HfO$_2$/Al$_2$O$_3$ multiple dielectric layer was measured on a Precision LCR Meter (E4980A, Agilent Technologies Inc., Santa Clara, CA, USA).

**Characterization of the magnetoelectrical performance of the IMOS devices.** The magnetoelectrical performance of the IMOS devices was characterized in a uniform magnetic field generated by a pair of Helmholtz coils. The field was swept and calibrated by a Hall sensor. The device was positioned in the field with different orientations as illustrated in Supplementary Fig. 7. The gate lines of the active matrix were driven by 3 eight-stage commercial shift registers soldered in series on the PCB (Supplementary Fig. 12). Acquisition of the electric signal was done through 8 channels of differential readout with 1 kHz sampling rate and 24-bit resolution ADC on the same PCB, which gave a frame rate up to 24 Hz for the 8 × 8 pixel array. The PCB with a microcontroller on board was connected to a computer via a universal serial bus (USB) interface, and controlled by a LabView program. The planar sensor arrays were also characterized with the same method to compare the magnetoelectrical performance before and after the self-folding process.

**Static mapping and dynamic tracking of magnetic objects.** The static mapping of magnetic stray field by the IMOS device was demonstrated by putting a rod NdFeB permanent magnet (1 mm diameter, 1 mm height) with axial magnetization over the sensor array, with a distance around 5 mm perpendicular to the bottom XZ plane. The differential voltages for each sensor plane were readout with an average of 10 sampling. The signals from dysfunctional pixels were interpolated and the obtained voltage maps were converted to the field strength maps by taking advantage of the sensitivity of each working pixel. The dynamic tracking of the motion of magnetic objects was performed by moving a diametrically magnetized NdFeB permanent magnet (9.5 mm diameter, 3 mm height) which were attached to a rotating stepper motor.

**Hair-embedded e-skin for mechanoreception.** The IMOS device was encapsulated within an epoxy layer to protect the sensor planes and facilitate further integration. A mixture solution of epoxy resin and hardener (EpoxAmite 103/2, KauPo Plankenhorn e. k. Spaichingen, Germany) with weight ratio of 100:28 was sandwiched between the sensor surface and a cover glass with a space of 500 μm. Then, the mixture was polymerized overnight in an oven at 40 °C to form a solid encapsulation. A magnetic hair is fabricated by adhering a small NdFeB magnet (1 mm diameter, 1 mm height) to a polymeric hair (shaft) and encapsulating both in epoxy, forming a magnetic bulb at the end of the hair. The hairs are punctured in a 2-mm-thick, ultra-flexible Ecoflex (Ecoflex 0010, Smooth-On, Inc., Eston, PA, USA) layer and placed on top of a molded PDMS spacer layer with cavities, so that the bulb of the magnetic hair is positioned in the cavities enabling them to move freely with a high degree of freedom. PDMS (Sylgard 184, Dow Corning) film with cavities was replicated from a laser-cut poly(methyl methacrylate) (PMMA) mold, and used as a spacer to interface between the Ecoflex membrane with hair-attached magnets and the IMOS device. The hairs were gently touched and the electric signal was recorded over time.

**Simulations.** The finite elements simulation results obtained from Ansys (ANSYS Academics 17.2 Electromagnetics) include the X-, Y- and Z- components of the stray magnetic field at several points in a square shaped planar area with the size of the actual IMOS device. The simulated grid has a spacing of 0.1 mm in X- and Z-direction, resulting in a total number of 88 × 88 simulated points, which leads to a higher spatial resolution compared to the maps obtained from the real IMOS device (8 × 8 pixels). The three magnetic field vector components X, Y and Z were transformed to reveal the projections on the actual sensitivity axes of the magnetic sensors. The transformation process is described in more detail in the Supplementary Information (Supplementary Note 2 and Supplementary Fig. 14).

## Data availability
Source data are provided with this paper.

## Code availability
Custom-developed LabView programs for the data collection are available from the corresponding author upon reasonable request.

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

## Acknowledgements

We thank M. Bauer (Leibniz IFW Dresden) for the help in the clean room. We thank C. Krien and I. Fiering (Leibniz IFW Dresden) for the deposition of metallic thin films. We thank K. Leger (Leibniz IFW Dresden) for the polymer syntheses. The support in the development of the experimental setups from the research technology department of the Leibniz IFW Dresden and the clean room team headed by R. Engelhard (Leibniz IFW Dresden) are greatly appreciated. We thank U. Nitzsche (Leibniz IFW Dresden) for the help with the simulation cluster. O.G.S. acknowledges support by the German Research Foundation DFG (Gottfried Wilhelm Leibniz Prize granted in 2018, SCHM 1298/22-1). D.K. acknowledges support by the German Research Foundation DFG (KA5051/1-1 and KA 5051/3-1), as well as by the Leibniz association (Leibniz Transfer Program T62/2019).

## Author contributions
C. B. and B. B. contributed equally to this work. C. B., B. B., D. K. and O. G. S. conceived the idea. C. B., B. B. and D. K. designed the experiment. C. B., B. B., D. D. K., V. K. B., B. R., Z. L., M. F. and D. K. performed the experiments. D. D. K. and C. B. performed the simulations. C. B., B. B. and D. K. analyzed the data. C. B., B. B. and D. K. wrote the manuscript with input from all authors. B. B., D. K. and O. G. S. supervised the work. All authors participated in the discussions.

## Funding

## Competing interests
The authors declare no competing interests.
