## [Peer Review File · Nature Communications]

Peer review comments, first round-

Reviewer #1 (Remarks to the Author):

Dear Authors,

In this manuscript, the interesting micro-origami architecture is demonstrated for 3D sensing of magnetic field. IGZO and Py were fabricated in wafer scale by sputtering, indicating compatible to various types of sensors and devices. The manuscript is well organized with detail explanation for fabrication of each layers and for integration of layers in Fig. 1C. Characterization of basic transport properties is well discussed in Fig. 2. In Fig. 3 and 4, the demonstration of 3D sensing of magnetic field is presented. Overall, based on the interesting concept, the device operation is well demonstrated.

It would be great that following comments are helpful to improve the manuscript.

1, In Fig. 2f and 2i, the yield of device performance of IGZO-FETs and magnetic sensors. I think the mobility of IGZO-FET is reproduced with small distribution. However, the sensitivities in Fig. 2i seems broad distribution. Could the authors find reason for this broad distribution or the relation of 64 devices in the position in wafer? During fabrication, are there inhomogeneous thickness or composition? If the authors find reasons, it would be great to add the discussion in page 11.

2, The size of Fig. 4a,b and supplemental Fig. 11 is not addressed. I recommend adding the scale bar in the figures.

Reviewer #2 (Remarks to the Author):

In this paper, Becker et al. demonstrated 3D magnetic sensor arrays based on micro-origami technology. Fabricated by planner photolithography, one sensor pixel consists of three anisotropic magnetoresistance (AMR) sensors, while two of the sensors fold up actuated by the swelling of hydrogels deposited on the hinge. An 8 x 8 pixel array was realized and actuated by active matrix technique. The subpixel sensors can be scanned serially and the output voltage can be parallelly read out, allowing a fast response of the magnetic signals and realizing real-time dynamic tracking of magnetic objects. This novel approach of building 3D magnetic sensor arrays is a great application for self-folding micro-origami technology and very impressive. This paper represents a brand-new direction of magneto sensors and could promote many applications in smart devices and flexible electronics. The reviewer suggests that the manuscript can be published after the authors properly addressing the issues listed below.

1. For the 3D magnetic sensors, it is critical that all the vertical AMR sensors in the pixel array, i.e. SYZ and SXY, have the same 90-degree folding angle (actuated by the swelling hydrogel). However, characterization and analysis of this folding process are missing in the manuscript. What is the distribution of the folding angle in each of the sensors in the 8x8 pixels?

2. Each of the sensors (SXY, SXZ, and SYZ) measures the magnetic field along a sensitive direction, as indicated by the red, blue and green arrows in the Figure 2h. However, even though the sensor panels are orthogonal to each other, the three field-sensitive directions (and thus the measured magnetic fields) are not. The reviewer believes the three sensitive directions rest on the same plane.

Assuming the width of the panel is a , then this plane is defined by three points $[(0, 0, 0), (a, a, 0)$ and $(0, a, a)]$. Therefore, even though the device is 3D, the measured three magnetic signals are still 2D and on the same plane $[(0, 0, 0), (a, a, 0)$ and $(0, a, a)]$. The reviewer authors should clarify this in the manuscript with a proper comment, so that the readers can properly assess this work.

3. Magnetic hair embedded e-skin is demonstrated as a potential application for the 3D magnetic sensor arrays. However, it is not clear why tracking the motion of a magnet (or a magnetic hair) requires 3D magnetic sensors. In fact, two planar AMR sensors with sensitive axes perpendicular to each other can possibly do this job just fine. Therefore, it seems that this demonstration doesn't reveal the full potential of these 3D magnetic sensors. What are the possible applications for the 3D magnetic sensors, which can not be achieved by planar magnetic sensors?

Reviewer #3 (Remarks to the Author):

The paper reports on the design and development of a novel magnetic sensor device suitable for use in so-called e-skins. From the point-of-view of my expertise the interesting innovation is the development of integrated sensors at the micro scale that have been folded in such a way as to decode a 3D magnetic field. The authors demonstrate that these 3D pixels perform as desired with little degradation in performance compared with the pre-folded state. Using microfabrication approaches the cube-like, post-folded pixels are directly integrated with the device.

The paper is well-written and structured with clear objectives, methods and device evaluation. I especially appreciated the large figures and detailed discussion. I have a few comments for the authors.

1 - While I appreciate the detail at times it seemed more like a specification set and less a discussion of the overall concept associated design decision discussions in a way that could benefit other researchers.

2 - There is some discussion of the importance of stability in the folded sensors during fabrication. I would appreciate something about stability during operation and its effect on performance.

3 - In discussion on the self-folding process the authors state the mechanisms were allowed to self-fold until the planes were orthogonal. I do not see anything on how that was established or the variability from sensor to sensor.

Overall I found the paper to be a valuable contribution to the field and an innovation that others will want to understand.

REVIEWER COMMENTS

Reviewer #1 (Remarks to the Author):

Dear Authors,

In this manuscript, the interesting micro-origami architecture is demonstrated for 3D sensing of magnetic field. IGZO and Py were fabricated in wafer scale by sputtering, indicating compatible to various types of sensors and devices. The manuscript is well organized with detail explanation for fabrication of each layers and for integration of layers in Fig. 1C. Characterization of basic transport properties is well discussed in Fig. 2. In Fig. 3 and 4, the demonstration of 3D sensing of magnetic field is presented. Overall, based on the interesting concept, the device operation is well demonstrated. It would be great that following comments are helpful to improve the manuscript.

Reply: The authors thank the Reviewer for the comments and helpful suggestions.

1, In Fig. 2f and 2i, the yield of device performance of IGZO-FETs and magnetic sensors. I think the mobility of IGZO-FET is reproduced with small distribution. However, the sensitivities in Fig. 2i seems broad distribution. Could the authors find reason for this broad distribution or the relation of 64 devices in the position in wafer? During fabrication, are there inhomogeneous thickness or composition? If the authors find reasons, it would be great to add the discussion in page 11.

Reply: The authors thank the Reviewer for the comments. Indeed, as the Reviewer pointed out, the mobility of the a-IGZO TFTs has a small statistical distribution, but the sensitivity distributions for the sensors are relatively large. In our demonstration, the sensors have an effective area of $1.1 \times 1.1 \text{ mm}^2$ and the sputtering target sizes for the Py and IrMn are four inches which are much larger than the sensor area. Therefore, the sensitivity distribution is probably not from the sputtering nonuniformity. In Supplementary Figure 8 of our original submission (Supplementary Figure 9 in the revised manuscript), the spatial distributions of the sensitivities for the S_{XY} , S_{XZ} and S_{YZ} sensors also do not show a clear trend depending on the sensor positions.

As we normalized the sensitivities to the power supply voltage of 3.3 V for the pixels, the real voltage drops through the Wheatstone bridge sensors are much smaller than this value due to the resistance of the wires and vias, as well as the transistor channels in the on state. We speculate that the relatively broad distribution of sensitivity is mainly from the variation of the resistance of the vias connecting the gate and the source-drain layers (Figure R1). In our fabrication, the dielectric layer is dry-etched and there are sharp transitions on the edges of the vias from the gate layer to the source-drain layer (Vias 1 to 8 in Figure R1b). The resistance of these vias may have a small variation depending on the coverage of the sharp edges by the sputtered source-drain metal layer. Therefore, we see some variations of the sensitivities between pixels from different source-drain lines (Supplementary Figure 8 in our original submission).

To improve the wire conductivity, we already used Cr/Au/Cr as the gate layer, Ti/Au as the source-drain layer, and Au as the sensor interconnecting layer in our experiments. In the future, via filling technologies will be adopted, for instance, depositing additional metal layers to decrease the variation of the via resistance and further improve the uniformity of the sensitivities.

We have added the discussion on the sensitivity distribution in the revised manuscript. Please see: Page 11, Line 14-18.

Figure R1. Layout design of the sensor device and the vias. **a**, Layout design of a device. **b**, Vias from the gate layer to the source-drain layer. **c**, Vias from the source-drain layer to the sensor layer.

2, The size of Fig. 4a,b and supplemental Fig. 11 is not addressed. I recommend adding the scale bar in the figures.

Reply: The authors thank the Reviewer for the suggestion. The scale bars are added. Please see: Page 16, Figure 4; Page 39, Supplementary Figure 12.

Reviewer #2 (Remarks to the Author):

In this paper, Becker et al. demonstrated 3D magnetic sensor arrays based on micro-origami technology. Fabricated by planner photolithography, one sensor pixel consists of three anisotropic magnetoresistance (AMR) sensors, while two of the sensors fold up actuated by the swelling of hydrogels deposited on the hinge. An 8 x 8 pixel array was realized and actuated by active matrix technique. The subpixel sensors can be scanned serially and the output voltage can be parallelly read out, allowing a fast response of the magnetic signals and realizing real-time dynamic tracking of magnetic objects. This novel approach of building 3D magnetic sensor arrays is a great application for self-folding micro-origami technology and very impressive. This paper represents a brand-new direction of magneto sensors and could promote many applications in smart devices and flexible electronics. The reviewer suggests that the manuscript can be published after the authors properly addressing the issues listed below.

Reply: The authors thank the Reviewer for the positive comments and the helpful suggestions.

1. For the 3D magnetic sensors, it is critical that all the vertical AMR sensors in the pixel array, i.e. SYZ and SXY, have the same 90-degree folding angle (actuated by the swelling hydrogel). However, characterization and analysis of this folding process are missing in the manuscript. What is the distribution of the folding angle in each of the sensors in the 8x8 pixels?

Reply: The authors thank the Reviewer for the helpful comment. According to the Reviewer's suggestion, we did more experiments to characterize the folding process and benchmark the folding angles, as shown in Figure R2. The typical self-folding process of a single pixel is shown in Figure

R2a. Once the sacrificial layer (SL) underneath the XY and YZ planes are etched, these two planes are released from the substrates and gradually folded up being actuated by the swelling hydrogel (HG). The self-folding process is stopped when the XY and YZ planes are closed to touch supporting each other.

To benchmark the folding angles, we define the angle between XY and YZ as θ_1 , the angle between YZ and XZ as θ_2 , and the angle between XY and XZ as θ_3 (Figure R2b). θ_1 is directly measured from the top view image of the cubic sensor (Figure R2c). θ_2 and θ_3 are calculated based on simple trigonometric relation as shown in Figure R3. The spatial and statistical distributions of the folding angles for a typical sensor device with 8×8 pixels are shown in Figure R2d-f. The folding yield for this device is 93.8%. The average values for θ_1 , θ_2 and θ_3 are $(90.4 \pm 0.6)^\circ$, $(87.1 \pm 1.7)^\circ$ and $(93.3 \pm 1.1)^\circ$, respectively. The small deviations of θ_2 and θ_3 from 90° are possibly from the obstruction of the hinges, which can be further improved by decreasing the hinge size. From the distributions of the folding angles we can see that the orthogonal self-folding process is highly controllable and reproducible from pixel to pixel.

a Self-folding process of a single pixel

b Illustration of the folding angles

c Top view micrographs of the self-folded cubic sensor pixel

d Distribution of folding angle θ_1

e Distribution of folding angle θ_2

f Distribution of folding angle θ_3

Figure R2. Self-folding process and distributions of the folding angles. **a**, Image series showing the self-folding process of a single pixel. Scale bar, $200 \mu\text{m}$. **b**, Illustration of the angles between different planes of the folded cubes. **c**, Top view micrographs of a folded sensor pixel showing the

measurements of the angle between YZ and XY planes, and the projected distance between the edges of YZ and XZ planes. Scale bars, 100 μm . **d-f**, Spatial and statistical distributions of the folding angles. The failed pixels are marked by gray color.

Figure R3. Calculation of θ_2 and θ_3 . **a**, Top view micrograph of a folded sensor pixel with a focus on the top edge of the XY and YZ planes. **b**, Top view micrograph of a folded sensor pixel with a focus on the bottom substrate. **c**, Illustration showing the calculation of the θ_2 and θ_3 angles.

We have added the discussion on the folding process and the distribution of the folding angles in the revised manuscript. Please see: Page 5, Line 8-18; Page 29, Supplementary Figure 1.

2. Each of the sensors (SXY, SXZ, and SYZ) measures the magnetic field along a sensitive direction, as indicated by the red, blue and green arrows in the Figure 2h. However, even though the sensor panels are orthogonal to each other, the three field-sensitive directions (and thus the measured magnetic fields) are not. The reviewer believes the three sensitive directions rest on the same plane. Assuming the width of the panel is a , then this plane is defined by three points $[(0, 0, 0), (a, a, 0)$ and $(0, a, a)]$. Therefore, even though the device is 3D, the measured three magnetic signals are still 2D and on the same plane $[(0, 0, 0), (a, a, 0)$ and $(0, a, a)]$. The reviewer authors should clarify this in the manuscript with a proper comment, so that the readers can properly assess this work.

Reply: The authors appreciate the reviewer for the valuable comment. Indeed, in our specific sensor magnetization case each single pixel has a virtual sensitivity plane which is similar to the 2D case, and tilted to the substrate plane. This configuration is not critical for the proof-of-concept demonstration elaborated in our work. By adjusting the initial magnetization directions during the fabrication, 3D magnetic sensors with different configurations can be achieved. In this work, we intended to emphasize on the single-step deposition and magnetization of the magnetic layer. However, by performing multiple (two or three) deposition and magnetization steps, 2D magnetic layers with different sensitivity orientations can be fabricated. Subsequent folding will lead to the orthogonal sensitivity directions that cannot be achieved for planar 2D cases, even with multiple depositions. Normally this would require 3D structuring of the wafer surfaces with all of the associated technological and performance complications.

According to the Reviewer's suggestion, we have added this discussion and clarification in the revised manuscript. Please see: Page 17, Line 15-29; Page 40-41, Supplementary Note 1 and Supplementary Figure 13.

3. Magnetic hair embedded e-skin is demonstrated as a potential application for the 3D magnetic sensor arrays. However, it is not clear why tracking the motion of a magnet (or a magnetic hair) requires 3D magnetic sensors. In fact, two planar AMR sensors with sensitive axes perpendicular to each other can possibly do this job just fine. Therefore, it seems that this demonstration doesn't reveal the full potential of these 3D magnetic sensors. What are the possible applications for the 3D magnetic sensors, which can not be achieved by planar magnetic sensors?

Reply: The authors thank the Reviewer for the comment. In our demonstration, a 3D magnetic sensor array is used to detect stray field distributions of multiple magnets that are not precisely fixed in place within the soft skin layer. These magnets are allowed to move within the skin layer in various possible directions. The 3D magnetic sensor array could enable the detection of the mechanical motions of the hairs in different directions which would be hard to realize with pairs of planar AMR sensors per hair. We emphasized on this point in the revised manuscript. Please see: Page 3, Line 26-27; Page 3, Line 28 - Page 4, Line 2; Page 15, Line 28 - Page 16, Line 3.

Reviewer #3 (Remarks to the Author):

The paper reports on the design and development of a novel magnetic sensor device suitable for use in so-called e-skins. From the point-of-view of my expertise the interesting innovation is the development of integrated sensors at the micro scale that have been folded in such a way as to decode a 3D magnetic field. The authors demonstrate that these 3D pixels perform as desired with little degradation in performance compared with the pre-folded state. Using microfabrication approaches the cube-like, post-folded pixels are directly integrated with the device. The paper is well-written and structured with clear objectives, methods and device evaluation. I especially appreciated the large figures and detailed discussion. I have a few comments for the authors.

Reply: The authors thank the Reviewer for the comment.

1 - While I appreciate the detail at times it seemed more like a specification set and less a discussion of the overall concept associated design decision discussions in a way that could benefit other researchers.

Reply: The authors thank the Reviewer for the comment. Our design decision on the self-folding sensors was based on three main factors: 1) The allocation of 3D reshaped magneto-sensory pixels within an array and integration of the a-IGZO TFT active matrix allow to detect the 3D distributed magnetic stray fields of loosely fixed magnets. These magnets are embedded within the soft elastomeric skin layer which can move within this elastomeric layer and therefore the matrix approach simultaneously provides means for detecting these spatially distributed stray fields and a convenient information readout. 2) The planar planes on the cube sides provide proper locations and area for deploying and reorientation of the sensing elements and ensuring precise alignment in 3D space. 3)The 3D architecture can be microfabricated through the precise planar 2D process.

The authors extended the discussion on these considerations in the manuscript. Please see: Page 3, Line 28 - Page 4, Line 2; Page 5, Line 22-26.

2 - There is some discussion of the importance of stability in the folded sensors during fabrication. I would appreciate something about stability during operation and its effect on performance.

Reply: The authors thank the Reviewer for the comment. According to the Reviewer's suggestion, we carried out additional experiments to test the operational stabilities of the a-IGZO TFTs and the integrated sensors, as shown in Figure R4. First, constant-voltage-bias-stress test was performed to investigate the stability of the a-IGZO TFTs (Figure R4a, b). From Figure R4a we can see that during the stress over a duration of 3600 s, the drain current drops within one order of magnitude, and the leakage current dose not increase. From the transfer curves before and after the stress test we can see that the a-IGZO TFT is still functioning well after the stress process (Figure R4b). Then, the dynamic stability of the signals from the IMOS sensors was tested by continuously recording the output voltages for 3 min. During the recording, a small magnet was swiped over the IMOS device to stimulate some spike signals. As shown in Figure R4c, the output signals show superb stability with flat baselines over 3 min. The signal drifting for all the three sensors in the pixel are very small. The results indicate that the IMOS devices are with high operational stability.

Figure R4. Operational stabilities of the a-IGZO TFTs and the IMOS sensors. **a, b**, Constant-voltage-bias-stress test for the a-IGZO TFTs. **a**, Drain current and gate leakage current change as a function of stress time. Over a stress duration of 3600 s with $V_{GS} = 3.3$ V, $V_{DS} = 1.0$ V, the drain current drops within one order of magnitude, and the leakage current dose not increase. **b**, Transfer curves before and after the constant-voltage-bias-stress test over a duration of 3600 s. The constant-voltage-bias stressing results in a positive transfer curve shift and a small decrease of the on-current. **c**, Dynamic stability of the output signal for a typical sensor pixel in the IMOS device. During a continuous recording over 3 min, the baselines for all the three subpixels are stable with very small signal drifting, indicating that the sensors are with high operational stability.

We have added the discussion on the stability during operation in the revised manuscript. Please see: Page 11, Line 23 to Page 12, Line 6; Page 37, Supplementary Figure 10.

3 - In discussion on the self-folding process the authors state the mechanisms were allowed to self-fold until the planes were orthogonal. I do not see anything on how that was established or the variability from sensor to sensor.

Reply: The authors thank the Reviewer for the comment. According to the Reviewer's suggestion, we did more experiments to characterize the self-folding process and the folding angles between different planes (Figure R2). For the detailed discussion please see the reply to the comment of Reviewer 2 (Question 1).

We have added the discussion on the self-folding process and the folding angles in our revised manuscript. Please see: Page 5, Line 8-18; Page 29, Supplementary Figure 1.

Overall I found the paper to be a valuable contribution to the field and an innovation that others will want to understand.

Reply: The authors thank again the Reviewer for the helpful suggestions and the recommendation.

Peer review comments, further round review-

Reviewer #1 (Remarks to the Author):

I appreciate the Authors for providing response and appropriate revisions.

Reviewer #2 (Remarks to the Author):

The revised article can be published at Nature Communication as is.

Reviewer #3 (Remarks to the Author):

I appreciated the response to my suggestions and questions. The additional materials that you have added to the paper are valuable.

REVIEWER COMMENTS

Reviewer #1 (Remarks to the Author):

I appreciate the Authors for providing response and appropriate revisions.

Reply: The authors would like to take this opportunity to thank the Reviewer for the helpful comments to improve the manuscript.

Reviewer #2 (Remarks to the Author):

The revised article can be published at Nature Communication as is.

Reply: The authors thank the Reviewer for the recommendation, and the helpful suggestions during the review process.

Reviewer #3 (Remarks to the Author):

I appreciated the response to my suggestions and questions. The additional materials that you have added to the paper are valuable.

Reply: The authors thank the Reviewer for the time to evaluate this manuscript, as well as all the helpful suggestions.